# USP52 regulates DNA end resection and chemosensitivity through removing inhibitory ubiquitination from CtIP

Ming Gao[1,2,3,4,6], Guijie Guo[1,2,6], Jinzhou Huang[1,2], Jake A. Kloeber [1,2,5], Fei Zhao[1,2], Min Deng[1,2], Xinyi Tu[1,2], Wootae Kim[1,2], Qin Zhou[1,2], Chao Zhang[1,2], Ping Yin[1,2], Kuntian Luo[1,2] & Zhenkun Lou [1,2✉]

Human C-terminal binding protein (CtBP)–interacting protein (CtIP) is a central regulator to initiate DNA end resection and homologous recombination (HR). Several studies have shown that post-translational modifications control the activity or expression of CtIP. However, it remains unclear whether and how cells restrain CtIP activity in unstressed cells and activate CtIP when needed. Here, we identify that USP52 directly interacts with and deubiquitinates CtIP, thereby promoting DNA end resection and HR. Mechanistically, USP52 removes the ubiquitination of CtIP to facilitate the phosphorylation and activation of CtIP at Thr-847. In addition, USP52 is phosphorylated by ATM at Ser-1003 after DNA damage, which enhances the catalytic activity of USP52. Furthermore, depletion of USP52 sensitizes cells to PARP inhibition in a CtIP-dependent manner in vitro and in vivo. Collectively, our findings reveal the key role of USP52 and the regulatory complexity of CtIP deubiquitination in DNA repair.

[1] Department of Molecular Pharmacology and Experimental Therapeutics, Mayo Clinic, Rochester, MN 55905, USA. [2] Department of Oncology, Mayo Clinic, Rochester, MN 55905, USA. [3] State Key Laboratory of Environmental Chemistry and Ecotoxicology, Research Center for Eco-Environmental Sciences, Chinese Academy of Sciences, Beijing 100085, China. [4] University of Chinese Academy of Sciences, Beijing 100049, China. [5] Mayo Clinic Medical Scientist Training Program, Mayo Clinic, Rochester, MN 55904, USA. [6]These authors contributed equally: Ming Gao, Guijie Guo. ✉email: lou.zhenkun@mayo.edu

D ouble-strand breaks (DSBs) are known to be one of the most lethal types of DNA lesions in mammalian cells[1,2]. Exogenous and endogenous DNA damaging agents such as radiation, carcinogens, and replication stress destroy the integrity and stability of genome, and contribute to the pathogenesis of various diseases, including developmental defects, immune deficiency, premature aging, and cancer[1,3–5]. To eliminate the risk of DSBs, cells have evolved a complex network to sense and repair damaged DNA, collectively known as the DNA damage response (DDR) pathway[5–7]. DSBs are typically repaired by the nonhomologous end-joining (NHEJ) pathway or the homologous recombination (HR) pathway. NHEJ is a highly error-prone DNA repair mechanism functioning throughout the cell cycle that directly ligates broken DNA ends in the absence of sequence homology. Conversely, HR is considered as an error-free mechanism which requires an intact sister chromatid as the template in S/G2 phases of the cell cycle[2,5,8].

DNA end resection initially generates 3′ single-stranded DNA (ssDNA) which provides a platform for recruiting HR repair-related proteins and prevents DNA repair by NHEJ[9,10]. C-terminal binding protein (CtBP) interacting protein (CtIP) is essential for the initiation of DNA end resection. CtIP acts as a short-range resection endonuclease and generally functions in association with the MRE11–RAD50–NBS1 (MRN) complex to determine DSB repair pathway choice[11,12]. After short, ssDNA is generated by CtIP/MRN complex, downstream nucleases such as exonuclease 1 (EXO1) or DNA replication helicase 2 and Bloom syndrome are further recruited to generate extended 3′-ssDNA for HR-mediated repair[11–13]. Besides its role in DNA end resection, CtIP also interacts with other proteins including retinoblastoma protein and breast cancer 1 (BRCA1) to regulate cell cycle progression in both a transcription dependent and independent manner[14,15]. In addition, CtIP knockout is embryonically lethal in mice and CtIP-depleted cells are hypersensitive to DNA damage induced by camptothecin, ionizing radiation (IR), and olaparib[16–19].

Ubiquitination is a dynamic and reversible process, including three enzymatic steps: E1 ubiquitin-activating enzymes, E2 ubiquitin-conjugating enzymes, and E3 ubiquitin ligases. Proteins can be conjugated with either a single ubiquitin (monoubiquitination), several ubiquitin molecules (multiubiquitination), or a chain of ubiquitins (polyubiquitination) to regulate diverse cellular processes[20]. The ubiquitination machinery has been reported to extensively orchestrate the DDR and maintains genome stability[21,22]. For example, RNF8/RNF168 pathway-dependent ubiquitination has an essential role in recruiting signaling and repair factors including 53BP1 and RAP80 to DSB sites[23,24]. In addition, RNF168 monoubiquitinates H2A/H2AX on K13–15 to control proper RNF8-mediated K63 chain extension and DSB signaling[23]. CtIP is ubiquitinated by APC/C[Cdh1] and RNF138 to regulate its stability and retention at DSB sites, respectively[25,26]. Deubiquitinases (DUBs) have been highlighted as key factors to dynamically oppose and reverse ubiquitin-mediated events. There are nearly 100 DUBs in the human genome and many of them play critical roles in orchestrating the DDR[27,28]. For instance, ubiquitin C-terminal hydrolase L3 (UCHL3) interacts with and deubiquitinates Rad51 to promote the interaction between Rad51 and BRCA2[29]; and USP13 regulates the formation of RAP80–BRCA1 complex foci formation through deubiquitinating RAP80[30]. Currently, the DUBs of CtIP have not been identified. Although USP4 was reported to interact with CtIP and promote CtIP recruitment to DSB sites, it showed no effect on CtIP ubiquitination[31,32]. Therefore, the mechanisms regulating CtIP deubiquitination remain unclear and need further investigation.

In this study, we perform a screen for DUBs of CtIP and find that USP52 specifically interacts with and deubiquitinates CtIP.

In addition, we describe the biological function and regulatory mechanism of USP52 in DNA end resection and CtIP-dependent HR following DNA damage. We also show that USP52 depletion renders cells more sensitive to PARP inhibition in vitro and in vivo. In conclusion, this study demonstrates a role for the USP52 DUB in regulating the activity of CtIP and provides a potential target for cancer therapy.

## Results

**USP52 interacts with and deubiquitinates CtIP.** CtIP is well-known to initiate DNA end resection and plays a critical role in DNA DSB repair[11]. Previous studies have shown that several E3 ligases or their substrate adapters, such as APC/C[Cdh1] and KLHL15 control CtIP ubiquitination and protein stability[17,25], however, the process of CtIP deubiquitination is still unknown. To study this, we overexpressed a panel of FLAG-tagged DUBs in HEK293T cells and performed co-immunoprecipitation (Co-IP) to screen for potential DUBs of CtIP. As shown in Supplementary Fig. 1a, among these DUBs, only USP4 and USP52 had observable interactions with CtIP. Because it was reported that USP4 interacts with CtIP but does not affect the latter's ubiquitination level[31], we chose USP52 as the candidate DUB for CtIP. Consistent with this, reciprocal IP of FLAG–CtIP in HEK293T cells brought down USP52 (Fig. 1a). In addition, we found that endogenous CtIP Co-IPed with endogenous USP52, and reciprocal IP of endogenous USP52 effectively pulled down CtIP (Fig. 1b, c). This further confirms the interaction between USP52 and CtIP in cells. Interestingly, the interaction between USP52 and CtIP did not change with IR treatment (Supplementary Fig. 1b), suggesting their interactions are not DNA damage inducible. We next mapped the binding region(s) between USP52 and CtIP using a series of USP52 and CtIP truncating deletions. As shown in Fig. 1d–g, the N-terminal region (AA17–AA160) of CtIP and the WD40 domain of USP52 are indispensable for the interaction between CtIP and USP52.

Since USP52 is a DUB, we next explored whether USP52 regulates CtIP protein level and ubiquitination. We first treated control cells or cells overexpressing USP52 with cycloheximide (CHX) to detect the protein level of CtIP. As shown in Supplementary Fig. 1c, CtIP protein level did not change after CHX treatment, indicating that USP52 does not affect the protein stability of CtIP. Next, we examined the effect of USP52 on the ubiquitination of CtIP. As shown in Fig. 1h, i and Supplementary Fig. 1d, e, depletion of USP52 increased conjugation of ubiquitin to CtIP about threefold compared to control, suggesting that USP52 regulates CtIP deubiquitination. A close examination of the size of ubiquitinated CtIP suggested that it was not extensively ubiquitinated, therefore suggesting that CtIP was not polyubiquitinated. To further strengthen our conclusion that CtIP is a substrate of USP52, we overexpressed WT USP52 or USP52 lacking its catalytic domain (ΔUCH). As shown in Fig. 1j, k and Supplementary Fig. 1f, g, re-expression of USP52 removed about 75% of ubiquitin from CtIP, whereas reconstitution of USP52 ΔUCH had almost no effect on CtIP ubiquitination. This allowed us to conclude that USP52 enzymatic activity is essential for regulating CtIP deubiquitination. In addition, we performed in vitro deubiquitination assays using ubiquitin-AMC (Ub-AMC), one of the most reliable artificial fluorescent DUB substrates, to detect the activity of USP52. As shown in Supplementary Fig. 1h, WT USP52 but not USP52 ΔUCH was able to cleave Ub-AMC and release free AMC fluorescence, indicating that the DUB activity of USP52 is directly responsible for hydrolyzing ubiquitin from its substrates. Moreover, we tested whether USP52 regulates CtIP ubiquitination after DNA damage. As shown in Fig. 1l, m and Supplementary Fig. 1i, j, about eighty

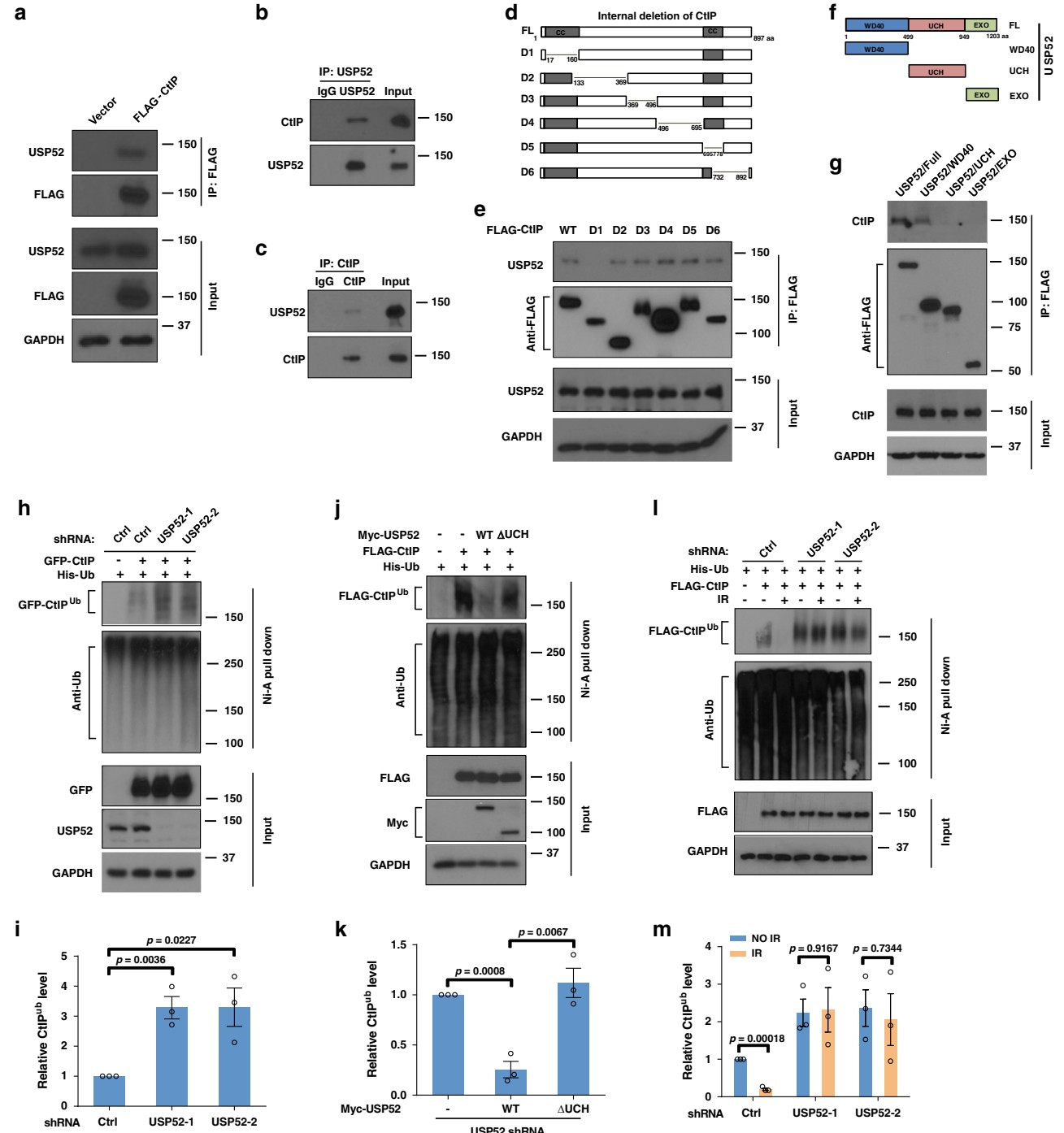

percent of the ubiquitination level of CtIP was reduced when treated with IR, suggesting a DNA damage-induced CtIP deubiquitination occurring in cells. However, USP52 depletion almost completely blocked this phenomenon in cells. Taken together, these results suggest that USP52 is a DUB that regulates CtIP deubiquitination under physiological conditions and in the context of DNA damage.

**USP52 promotes DNA end resection and HR.** Due to the important role of CtIP in DNA end resection and ssDNA generation, we hypothesized that USP52 might be involved in these processes through regulating the activities of CtIP. We first examined whether USP52 regulates the recruitment of CtIP onto DNA damage sites. As shown in Fig. 2a, b, CtIP focus formation

was significantly increased when treated with IR, but USP52 depletion had no effect on CtIP focus formation before or after IR treatment. In addition, we used a ChIP assay with the ER-AsiSI system to detect whether the DSB recruitment of CtIP is affected by USP52. In this assay, the restriction enzyme AsiSI translocates to the nucleus following 4-OHT treatment and generates DSBs at sequence-specific sites (Fig. 2c). As shown in Fig. 2d, the recruitment of FLAG–CtIP to DSBs was significantly increased when cells were treated with 4-OHT, but there was almost no difference in control and USP52-depleted cells, suggesting that USP52 does not affect the recruitment of CtIP to DSBs. Next, we performed a quantitative resection assay using the ER-AsiSI system to measure the effect of USP52 on ssDNA production. As shown in Fig. 2e, depletion of USP52 resulted in a significant

**Fig. 1 USP52 interacts with and deubiquitinates CtIP. a** HEK293T cells were transfected with vector or Flag–CtIP for 24 h, cells were then lysed and immunoprecipitated with anti-FLAG agarose beads. The beads were boiled and analyzed with indicated antibodies. **b, c** HEK293T cell lysates were subject to immunoprecipitation with control IgG, anti-USP52 (**b**) or CtIP antibodies (**c**). The immunoprecipitates were then blotted with indicated antibodies. **d, e** Schematic representation of CtIP constructs used in this study (**d**). HEK293T cells were transiently transfected with the indicated CtIP constructs for 24 h, then cell lysates were incubated with anti-FLAG agarose beads overnight at 4 °C. The immunoprecipitates were then blotted with indicated antibodies (**e**). **f, g** Schematic representation of USP52 constructs used in this study (**f**). Cellular lysates from HEK293T cells transfected with the indicated constructs of USP52 were immunoprecipitated with anti-FLAG agarose beads, and then western blot was performed with indicated antibodies (**g**). WD40 WD40 repeat domain, UCH ubiquitin C-terminal hydrolase domain, EXO exonuclease domain. **h, i** Control or USP52-depleted HEK293T cells were transfected with CtIP and His-Ub for 24 h. Harvested cells were subjected to immunoprecipitation using nickel (His) beads and then the level of ubiquitin conjugates of CtIP was detect by western blotting assay (**h**). The quantification of bands was analyzed by Image J and data are presented as mean values ± SEM from three independent experiments (**i**). **j, k** USP52-depleted HEK293T cells expressing the indicated USP52 constructs were harvested and then immunoprecipitated with nickel (His) beads. Blots were performed to detect the level of ubiquitin conjugates of CtIP (**j**). The quantification of bands was analyzed by Image J and data are presented as mean values ± SEM from three independent experiments (**k**). **l, m** Control or USP52-depleted HEK293T cells were transfected with CtIP and His-Ub for 24 h, cells were then untreated or treated with IR for 2 h. Cell lysates were immunoprecipitated with nickel (His) beads and then blots were performed to detect the level of ubiquitin conjugates of CtIP (**l**). The quantification of bands was analyzed by Image J and data are presented as mean values ± SEM from three independent experiments (**m**).

decrease of ssDNA levels at all distances from the DSB sites. These results suggest that USP52 regulates CtIP function in DNA end resection.

The ssDNA from end resection is quickly protected and coated with replication protein A (RPA) complex and then replaced by Rad51 to facilitate HR-based DNA repair[10,33]. Therefore, we continued to explore the role of USP52 in RPA2 and Rad51 focus formation. As shown in Fig. 2f–i, depletion of USP52 significantly decreased the formation of RPA and Rad51 foci. In addition, the intensities of RPA2 and Rad51 foci were decreased in USP52-depleted cells compared to control cells (Supplementary Fig. 2a, b), indicating that both the number and intensity of RPA2 and Rad51 foci were decreased when USP52 was depleted. We next examined the effect of USP52 on the two major DNA repair pathways and found that USP52 depletion decreased the efficiency of HR— but not NHEJ-based DNA repair (Fig. 2j and Supplementary Fig. 2c). In addition, because initiation of CtIP-dependent resection is also required for MMEJ, we examined the role of USP52 in MMEJ-based DNA repair. As shown in Supplementary Fig. 2d, MMEJ efficiency was also impaired in cells with USP52 depletion. Next, we explored whether USP52 affects DSB resection and HR through its interaction with CtIP. As shown in Supplementary Fig. 2e and Fig. 2k, l, WT USP52, but not the USP52 ΔWD40 truncate, successfully rescued DSB resection and HR in USP52-depleted cells, suggesting that the USP52–CtIP interaction is important for the role of USP52 in DSB resection and HR. Because ASF1A is known to be stabilized by USP52[34], we further investigated whether the effect of USP52 depletion on DSB resection and HR repair is partly due to decreased ASF1A expression. As shown in Supplementary Fig. 2f–h, ectopic expression of ASF1A could not rescue USP52 depletion-induced defects in DSB resection and HR efficacy, suggesting that ASF1A is not involved in USP52-mediated DSB resection and HR repair.

RPA–ssDNA complex is also essential for the phosphorylation of CHK1 and RPA to regulate the cellular sensitivity to DNA damage agents[10,33]. As expected, we found that USP52 depletion attenuated IR-induced CHK1 and RPA phosphorylation, and increased IR sensitivity (Supplementary Fig. 2i and Fig. 2m). Because PARP inhibitors (PARPi) are promising drugs for the treatment of tumors with impaired HR capability[7,35], we hypothesized that USP52 depletion might sensitize cells to PARPi. Indeed, knocking down USP52 sensitized cells to PARPi (Fig. 2n). Taken together, these results establish that USP52 is critical for DSB-induced resection and HR repair.

**USP52 regulates resection and HR in a catalytic activity- and CtIP-dependent manner.** To investigate whether the regulation

of DNA end resection and HR by USP52 is dependent on its catalytic activity, we reconstituted USP52-deficient cells with WT USP52 or USP52 ΔUCH. As shown in Fig. 3a–f and Supplementary Fig. 3a, USP52 depletion sharply decreased ssDNA production, formation of RPA2 and Rad51 foci, HR efficiency, and phosphorylation of CHK1 and RPA2 compared to control cells. Reconstitution of WT USP52 but not USP52 ΔUCH into USP52-deficient cells rescued these phenotypes, suggesting that the catalytic activity of USP52 is important for its role in regulating DNA end resection and HR.

To further confirm that the regulatory role of USP52 in DNA end resection and HR is dependent on CtIP, we depleted both CtIP and USP52 in cells. As shown in Fig. 3g, h and Supplementary Fig. 3b, knocking down USP52 did not further reduce ssDNA production, HR repair, or phosphorylation of CHK1 and RPA2 in CtIP-depleted cells, suggesting that USP52 regulates DNA end resection and HR in a CtIP-dependent manner.

Because USP52 does not affect CtIP level and localization at DNA damage sites, we further investigated how USP52 regulates CtIP function. CtIP dimerization is required for its localization to DSBs and function in DDR[36]. However, the dimerization of CtIP was unchanged when USP52 was depleted (Supplementary Fig. 3c). It was reported that the phosphorylation of CtIP at Ser-327 (S327) and Thr-847 (T847) are essential for its role in maintaining genomic stability[15,37,38]. The phosphorylation of CtIP at T847, which is critical for DNA end resection[37,38], was transiently increased between 0.5 and 2 h in response to IR (Supplementary Fig. 3d), consistent with another study[39]. We found that the phosphorylation of CtIP–S327, which is critical for BRCA1–CtIP interaction[15,33], was not affected by depletion of USP52 (Supplementary Fig. 3e). On the other hand, USP52 depletion significantly decreased IR-induced CtIP–T847 phosphorylation. Reconstituting WT USP52 but not USP52 ΔUCH into USP52-depleted cells effectively reversed the reduction of CtIP–T847 phosphorylation (Fig. 3i). Taken together, these results suggest that USP52 regulates end resection and HR at least partially through affecting CtIP–T847 phosphorylation. Because CtIP is phosphorylated at T847 mainly by CDKs[40], we asked whether USP52-regulated CtIP deubiquitination is also cell cycle dependent. To answer this, we examined the ubiquitination levels of CtIP at different stages of the cell cycle. As shown in Supplementary Fig. 3f, CtIP ubiquitination levels were comparable at different cell cycle phases in control cells and USP52-depleted cells, indicating that CtIP ubiquitination regulation by USP52 is not cell cycle dependent. In addition, we compared the ubiquitination levels of WT CtIP and CtIP–T847E mutant, which is able to overcome the cell cycle regulation of CtIP, and found

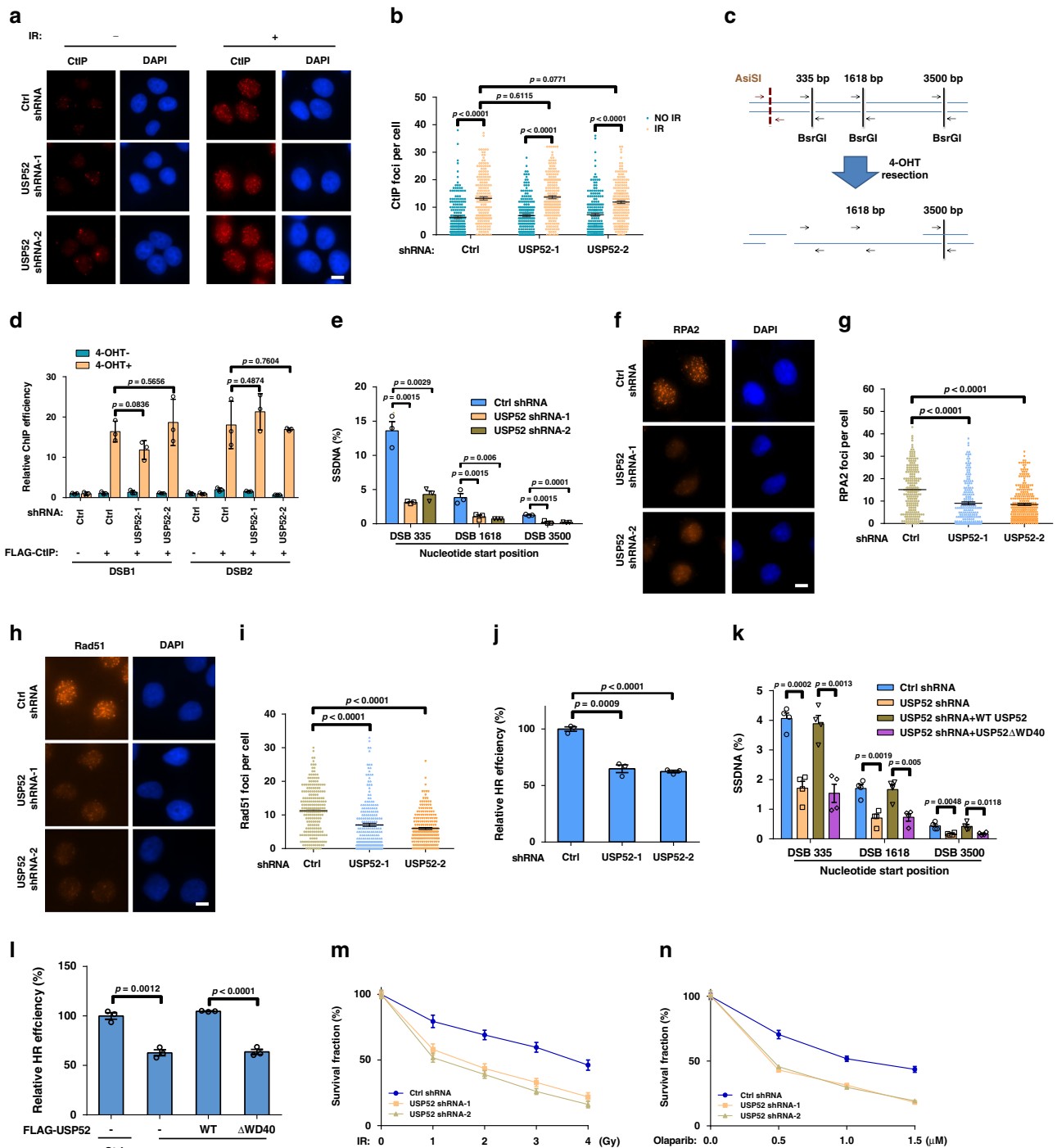

**Fig. 2 USP52 promotes DNA end resection and HR repair. a, b, f–i** Representative pictures (**a, f, h**) and quantifications (**b, g, i**) of CtIP (**a, b**), RPA2 (**f, g**) and Rad51 (**h, i**) foci in control or USP52-depleted U2OS cells when treated with 5 Gy IR for 3 h, 2 Gy IR for 2 h or 5 Gy IR for 5 h, respectively. Data are representative of three independent experiments. Each dot represents a single cell, and more than 200 cells were counted in each group for this experiment. Error bars represent SEM from this experiment. Scale bar, 10 μm. **c** Illustration of the designation of Taqman qPCR primers and probes (black arrows) for measuring resection at sites adjacent to the AsiSI sites (red arrows). The primer pairs are across BsrGI restriction sites. All Taqman probes are designed at either side of the restriction site. **d** Control or USP52-depleted ER-AsiSI U2OS cells transfected with FLAG–CtIP were added with or without 4-OHT. ChIP assay was then performed using FLAG antibody. **e** The genomic DNA extracted from control or USP52-depleted ER-AsiSI U2OS cells were digested with BsrGI. DNA end resection adjacent to DNA double-strand break sites was measured by qPCR assay. **j** The HR-mediated DSB repair efficacy of control and USP52 depletion in HEK293T cells were analyzed using HR reporter. Data are presented as mean values ± SEM from three independent experiments. **k** Control or USP52-depleted ER-AsiSI U2OS cells transfected with indicated constructs were used for DNA end resection assay. Data are presented as mean values ± SEM from four independent experiments. **l** The efficacy of HR in control or USP52-depleted HEK293T cells transfected with indicated USP52 constructs were analyzed using GFP reporter assay. **m, n** Control or USP52-depleted U2OS cells were exposed to the indicated dose of IR (**m**) or PARPi (**n**) for 2 weeks, colony formation assay was then performed to detect the survival rate of cells. **d, e, j, l–n** Data are presented as mean values ± SEM from three independent experiments.

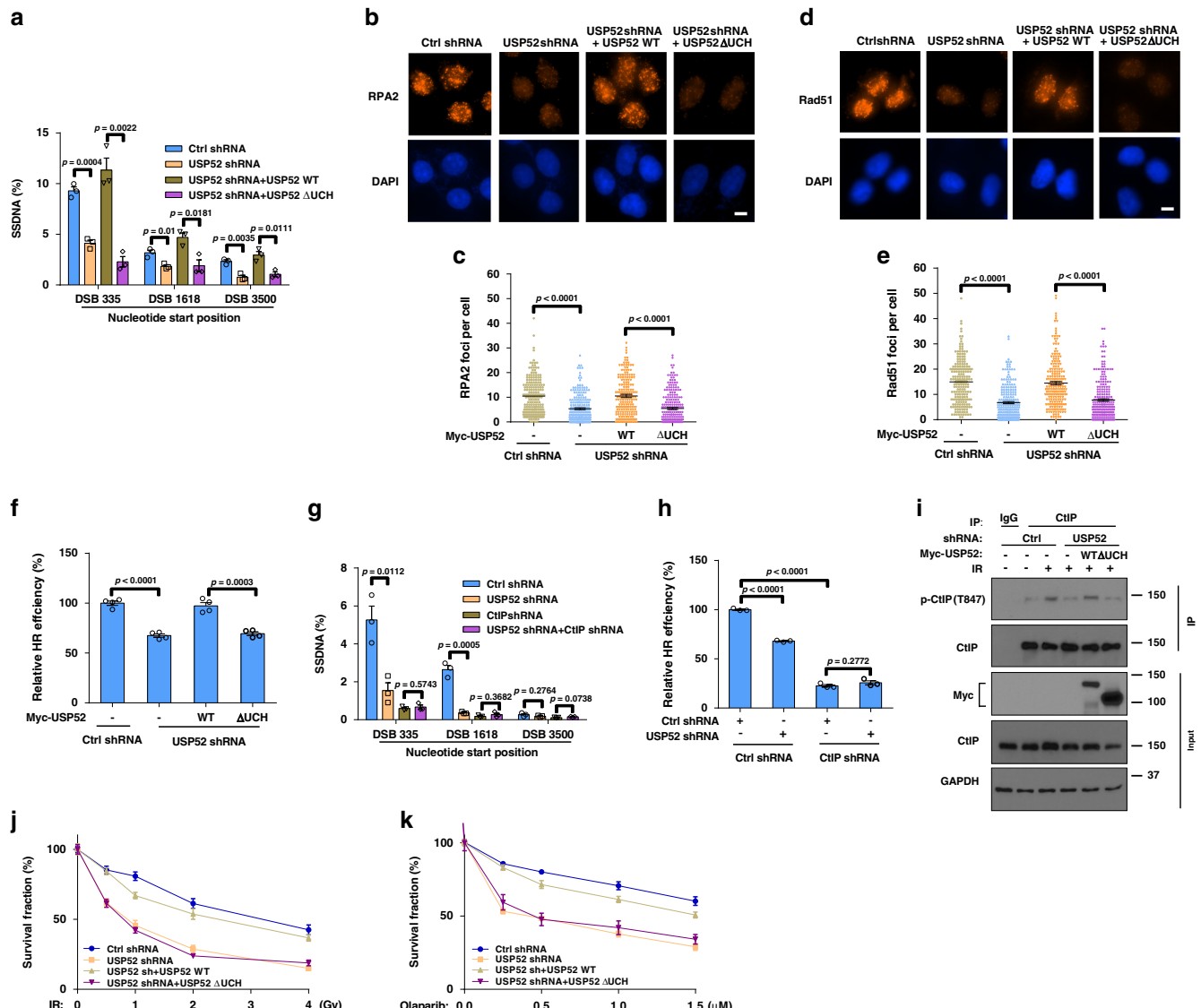

**Fig. 3 USP52 regulates DNA end resection and HR through CtIP. a** ER-AsiSI U2OS cells with or without USP52 depletion were transfected with indicated constructs for 24 h, then cells were treated with 1 μM 4-OHT for 4 h before genomic DNA was extracted and digested with BsrGI. DNA end resection efficiency was measured by qPCR assay. Data are presented as mean values ± SEM from three independent experiments. **b–e** Control or USP52-depleted U2OS cells were transfected with indicated constructs for 24 h before treated with 2 Gy IR for 2 h (**b, c**) or 5 Gy IR for 5 h (**d, e**). RPA2 or Rad51 focus formation was then detected by immunofluorescence (**b, d**) and quantified (**c, e**). Data are representative of three independent experiments. Each dot represents a single cell, and more than 200 cells were counted in each group for this experiment. Error bars represent SEM from this experiment. Scale bar, 10 μm. **f** Control or USP52-depleted HEK293T cells reconstituted with the indicated constructs and HR reporter were subjected to HR assay. Error bars represent SEM from three independent experiments. **g** Control or USP52-depleted ER-AsiSI U2OS cells were transfected with control or CtIP shRNA, and then cells were treated with 4-OHT for DNA end resection assay. Each bar represents SEM from three independent experiments. **h** Control or USP52-depleted HEK293T cells were transfected with control or CtIP shRNA and HR reporter, and then cells were harvested for HR assay. Error bars represent SEM from three independent experiments. **i** Control or USP52-depleted HEK293T cells were transfected with WT USP52 or USP52 ΔUCH for 24 h before treated with 5 Gy IR for 1 h. Cells were harvested and subject to immunoprecipitation with control IgG and CtIP antibodies. The immunoprecipitates were then blotted with indicated antibodies. **j, k** Control and USP52-depleted U2OS cells stably expressing WT USP52 or ΔUCH were in response to the indicated dose of IR (**j**) or PARPi (**k**) for 2 weeks. Cell viability was assessed using colony formation assay. Error bars represent SEM from three independent experiments.

that WT CtIP and CtIP–T847E mutant ubiquitination levels were both significantly decreased after IR treatment (Supplementary Fig. 3g). Moreover, CDK kinase inhibition by BMS-265246 had no effect on IR-induced CtIP deubiquitination (Supplementary Fig. 3h), further suggesting that CtIP ubiquitination/deubiquitination was not cell cycle dependent.

To further investigate whether USP52-mediated IR- and PARPi-hypersensitivity was also dependent on its catalytic activity, we reconstituted USP52-deficient cells with WT USP52

or USP52 ΔUCH to evaluate the sensitivity of cells to IR and PARPi. As shown in Fig. 3j, k, overexpression of WT USP52 but not USP52 ΔUCH reversed IR- or PARPi-hypersensitivity in USP52-depleted cells, indicating that USP52 regulates IR- and PARPi-hypersensitivity through its catalytic activity.

**Deubiquitination of CtIP by USP52 is important for resection and HR.** We further mapped the potential ubiquitination sites of

CtIP that are regulated by USP52. By using a series of CtIP deletion truncates (Fig. 1d) we found that the N-terminal and C-terminal domain of CtIP were indispensable for the ubiquitination of CtIP (Supplementary Fig. 4a). Previous mass spectrometric data showed that K62, K314, K360, k378, K410, K438, K526, K530, K585, K604, K613, K640, K760, and K782 were ubiquitination sites of CtIP by using UbiSite or SEPTM strategy[41,42]. Among them, K62, K760, and K782 were located within the N and C terminus of CtIP. Therefore, we generated mutants of candidate CtIP ubiquitination (KR) and performed ubiquitination assays. As shown in Supplementary Fig. 4b, two of the three single-site mutants (K760R and K782R) of CtIP had a significant decrease of ubiquitination level. Therefore, we mutated these two sites together and found that the double K to R mutant (2KR) almost totally abolished the ubiquitination on CtIP (Fig. 4a). These results indicate that these two lysines are the major ubiquitination sites on CtIP. Next, we explored whether CtIP ubiquitination affects end resection and HR. If USP52 regulates CtIP and HR through deubiquitinating CtIP, we predicted that the CtIP 2KR mutation would bypass the requirement of USP52. As shown in Fig. 4b–g, expression of CtIP 2KR but not WT CtIP rescued USP52 deficiency-induced defects in ssDNA production, RPA/Rad51 foci formation and HR repair. Because the K782R mutant of CtIP had a more significant decrease of ubiquitination levels compared to K760R mutant (Supplementary Fig. 4b), we further investigated whether the K782R mutant of CtIP alone was sufficient to compensate USP52 depletion-reduced resection and HR repair. As shown in Supplementary Fig. 4c, d, CtIP K782R mutant only partially restored USP52 depletion-reduced resection and HR repair, suggesting that deubiquitination of both K760 and K782 sites of CtIP are indispensable for the role of CtIP in resection and HR repair. In addition, expression of CtIP 2KR but not WT CtIP rescued defective CtIP–T847 phosphorylation in USP52-deficient cells (Fig. 4h). These results suggest that the ubiquitination of CtIP negatively regulates its phosphorylation at Thr-847 and biological functions; USP52 removes the inhibitory ubiquitination of CtIP to promote DNA end resection and HR following DNA damage.

We further investigated the biological outcome of CtIP ubiquitination in DDR. As shown in Supplementary Fig. 4e, expression of CtIP 2KR but not WT CtIP rescued USP52 deficiency-induced defects of CHK1 and RPA2 phosphorylation. In addition, CtIP 2KR but not WT CtIP was able to reverse increased IR- and PARPi-sensitivity caused by USP52 depletion (Fig. 4i, j), suggesting that deubiquitination of CtIP by USP52 is critical for repair of IR- and PARPi-induced DNA damage.

**Phosphorylation of USP52 by ATM under IR treatment**. The inducible deubiquitination of CtIP by USP52 suggests that USP52 itself might be regulated by the DNA damage signaling. We analyzed data from PhosphoSitePlus (https://www.phosphosite.org/homeAction.action) and found that USP52 has two putative pSQ/TQ motif sites (S469 and S1003), suggesting that USP52 might be a potential ATM/ATR substrate. Indeed, we found that USP52 was phosphorylated at SQ/TQ motifs in response to IR, and this phosphorylation could be abolished by pretreating cells with an ATM inhibitor or treating cell lysates with phosphatase (Fig. 5a), indicating that USP52 could be phosphorylated in an ATM-dependent manner following IR treatment. We next examined which SQ/TQ site of USP52 was phosphorylated and found that the S1003A but not S469A mutation mostly abolished the phosphorylation of USP52 after DNA damage (Fig. 5b and Supplementary Fig. 5a), suggesting that USP52 is phosphorylated on Ser-1003 in an ATM-dependent manner following IR treatment.

To further investigate the biological significance of USP52 phosphorylation, we compared the activities of WT USP52 and the S1003A mutant after DNA damage by in vitro deubiquitination assays. We purified WT USP52 or S1003A from cells before or after IR and then incubated with ubiquitinated CtIP. As shown in Fig. 5c, the DUB activity of WT USP52 increased significantly after IR treatment, whereas the activity of S1003A mutant did not change. These results suggest that phosphorylation of USP52 S1003 plays a vital role in regulating the DUB activity of USP52. To further analyze the role of USP52 phosphorylation in regulating DNA end resection and HR, we re-expressed WT USP52 and USP52 S1003A mutant in USP52-deficient cells to compare their biological functions. As shown in Fig. 5d–h, reconstitution of WT USP52, but not the S1003A mutant rescued USP52 depletion-induced defect in RPA/Rad51 foci formation and HR repair. To further examine whether the phosphorylation of USP52 at S1003 regulates CtIP deubiquitination, we reconstituted WT or the S1003A mutants in USP52-deficient cells and examined the ubiquitination level of CtIP with or without IR treatment. As shown in Fig. 5i, CtIP ubiquitination level was significantly decreased in cells expressing WT USP52 but not the S1003A mutant after IR treatment, suggesting that USP52 phosphorylation is critical for deubiquitinating CtIP following IR exposure. We hypothesized that ATM-mediated phosphorylation of USP52 activates its DUB activity. To test this, we performed an in vitro kinase reaction of USP52 by ATM followed by an in vitro deubiquitination assay. As shown in Fig. 5j, incubating WT, but not the S1003A mutant of USP52, with ATM further decreased the ubiquitination level of CtIP, suggesting that ATM-mediated USP52 phosphorylation is important for USP52 activation to deubiquitinate CtIP following DNA damage. In addition, CtIP T847 phosphorylation in USP52-deficient cells was reversed by WT USP52 but not the USP52 S1003A mutant in response to IR (Fig. 5k). These results suggest that USP52 phosphorylation by ATM is important for its role in regulating DNA end resection and HR through regulating CtIP deubiquitination.

We further evaluated whether USP52 phosphorylation by ATM affects CHK1 and RPA2 phosphorylation and render cells resistant to cancer therapy. As shown in Supplementary Fig. 5b, l, m, reconstitution of USP52-deficient cells with WT USP52 but not USP52 S1003A rescued USP52 depletion-induced defects, including CHK1 and RPA2 phosphorylation and IR and PARPi response, suggesting that USP52 phosphorylation by ATM is important for proper DDR.

**Targeting USP52-mediated CtIP deubiquitination sensitizes to PARPi in vivo**. To further strengthen the role of the USP52–CtIP pathway in HR and response to PARPi, we investigated the relationship between USP52 and CtIP in response to PARPi in vivo. As shown in Fig. 6a–c, USP52 depletion enhanced sensitivity to PARPi in an osteosarcoma xenograft model, which could be rescued by the overexpression of the 2KR mutant but not WT CtIP. These results indicate that the USP52–CtIP pathway is important for PARPi response. Therefore, the USP52–CtIP pathway may be a potential therapeutic target alongside PARPi or may serve as a biomarker of PARPi sensitivity.

## Discussion

CtIP is required for DNA end processing and essential for the initial step of homology-directed repair in eukaryotes[9,11]. Therefore, the protein level, the DNA damage site recruitment and the activity of CtIP are all tightly controlled for proper DNA end resection[17,38,43]. It was reported that post-translational modifications such as phosphorylation, sumoylation, and ubiquitination

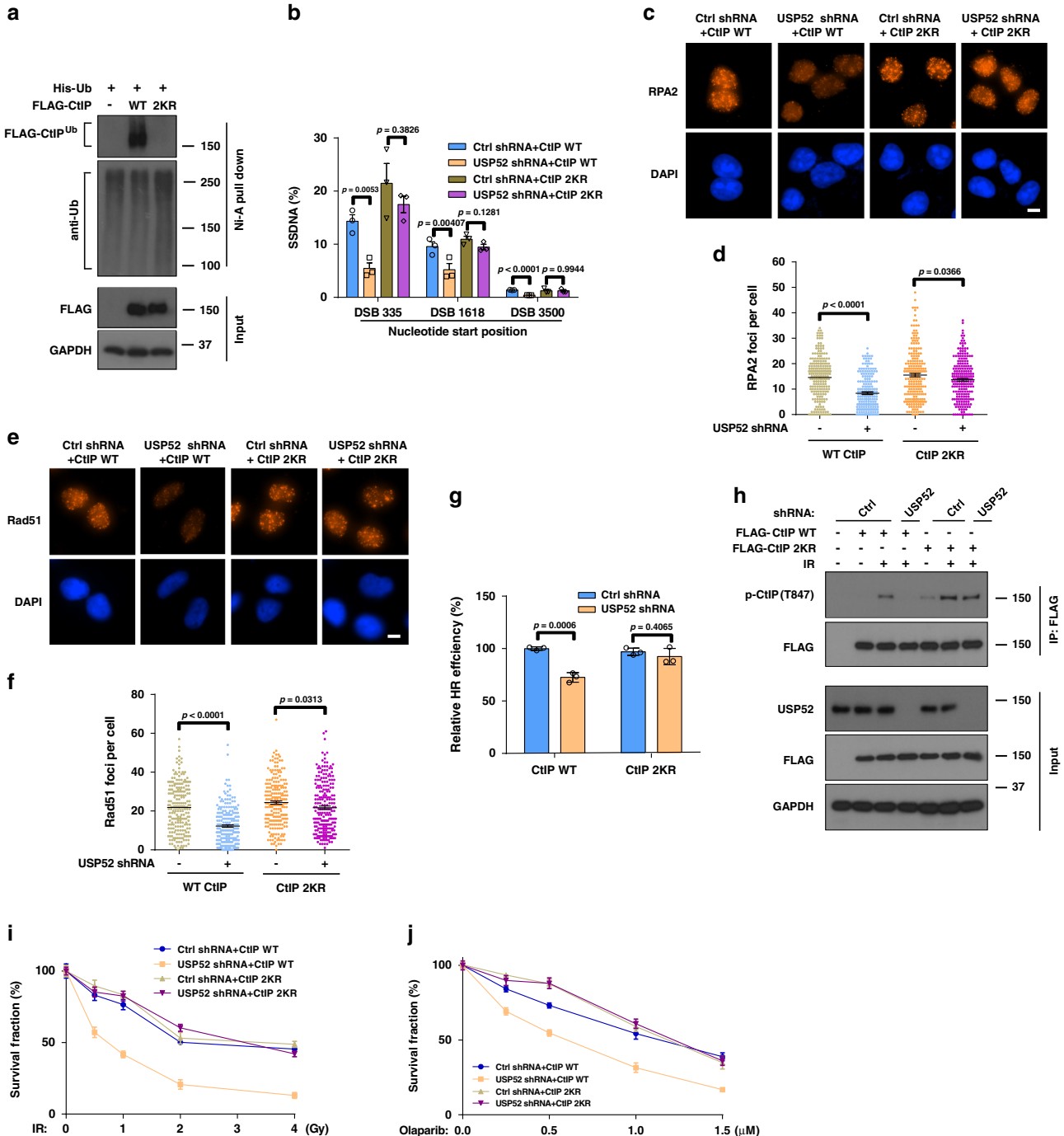

**Fig. 4 Deubiquitination of CtIP by USP52 is important for DNA end resection and HR. a** HEK293T cells were transfected with WT CtIP or CtIP 2KR for 24 h before harvesting and immunoprecipitation with nickel (His) beads. Blots were detected by indicated antibodies. **b** Control or USP52-depleted ER-AsiSI U2OS cells were transfected with WT CtIP or CtIP 2KR for 24 h, and then cells were treated with 4-OHT to induce DSB. Cells were harvested for DNA end resection analysis measured by qPCR assay. Each bar represents SEM from three independent experiments. **c–f** Representative images (**c**, **e**) and quantification (**d**, **f**) of RPA2 (**c**, **d**) and Rad51 foci (**e**, **f**) in control or USP52 knockdown U2OS cells which were transfected with WT CtIP or CtIP 2KR for 24 h before treated with 2 Gy IR for another 2 h or 5 Gy IR for 5 h. Data are representative of three independent experiments. Each dot represents a single cell, and more than 200 cells were counted in each group for this experiment. Error bars represent SEM from this experiment. Scale bar, 10 μm. **g** Control or USP52-depleted HEK293T cells which were transfected with WT CtIP or CtIP 2KR were subjected to DR-GFP-based HR assay. Data are presented as mean values ± SEM from three independent experiments. **h** Control or USP52-depleted HEK293T cells were transfected with WT or 2KR mutant of CtIP for 24 h prior to be treated with 5 Gy IR. Cells were then harvested and immunoprecipitated with anti-FLAG agarose beads to detect the phosphorylation of CtIP at T847. **i**, **j** The sensitivity of control or USP52-depleted cells stably expressing WT CtIP or 2KR truncates in response to IR (**i**) or PARPi (**j**) were analyzed by colony formation assay. Error bars represent SEM from three independent experiments.

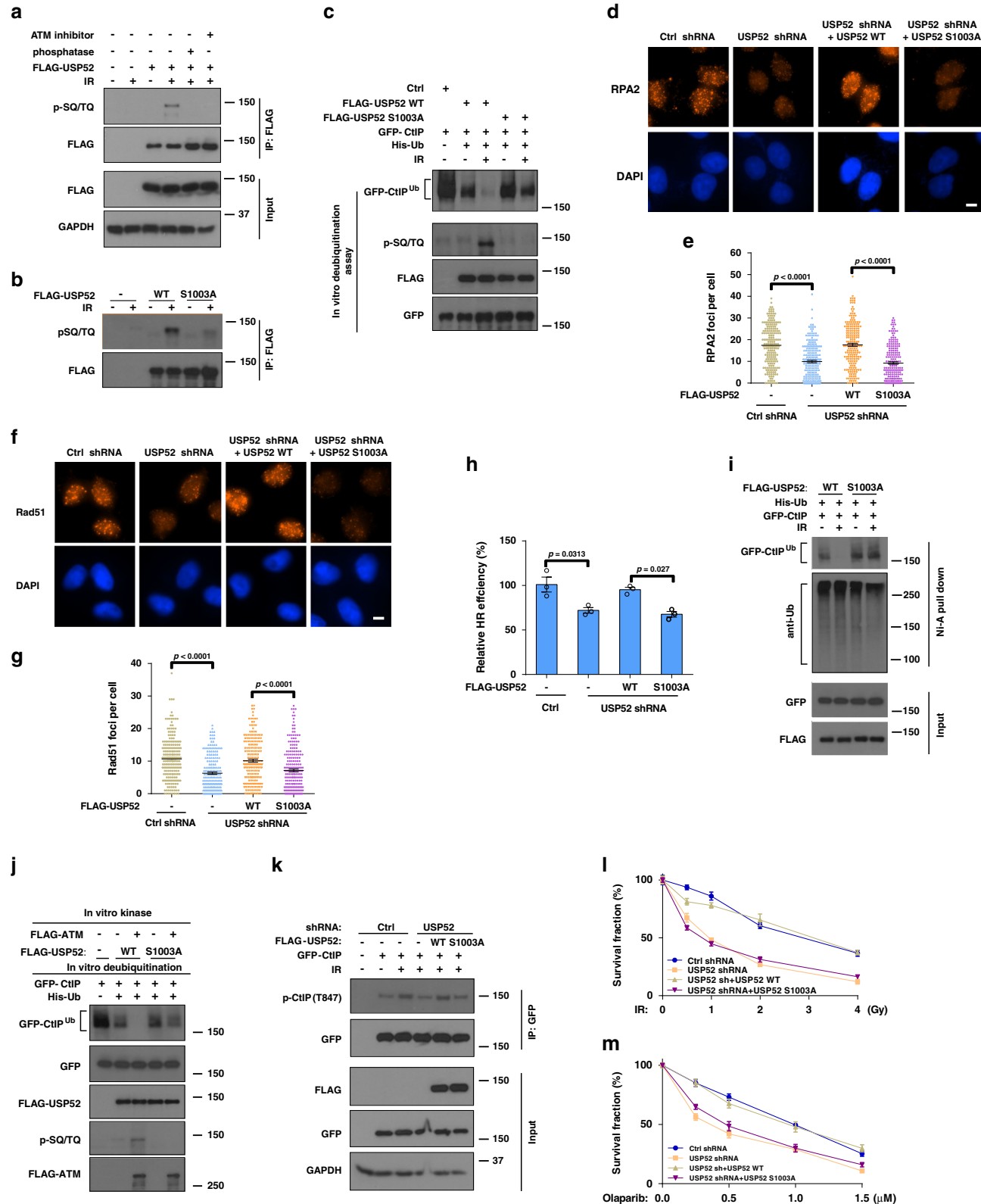

play important roles in CtIP regulation. For example, it is clear that CDK-mediated phosphorylation of CtIP on Thr-847 modulates ssDNA generation, RPA recruitment, and phosphorylation to ensure appropriate DNA end resection during S/G2 cell phase[37,38,40]. In addition, CDK-phosphorylated CtIP on Ser-327 promotes the interaction with BRCA1 and facilitates the ubiquitination of CtIP by BRCA1, which is required for CtIP

participated in G2/M checkpoint control[15]. Sumoylation of CtIP at lysine 896 by SUMO E3 ligase CBX4 is also important for the role of CtIP in regulating DNA end processing and genomic stability[43]. Several studies have reported that E3 ubiquitin ligases or their substrate adapter such as APC/C$^{Cdh1}$ and KLHL15 participate in regulating the protein stability and activity of CtIP[17,25]. Whether the ubiquitin on CtIP protein affects its function and

**Fig. 5 ATM kinase promotes the activity of USP52 to regulate the DDR. a** HEK293T cells transfected with FLAG–USP52 were treated with DMSO or 25 μM Ku55933 for 2 h prior to IR treatment. Harvested cells were immunoprecipitated with anti-FLAG agarose beads. After untreated or treated with lambda protein phosphatase, blots were probed with pSQ/TQ antibody. **b** HEK293T cells transfected with indicated USP52 constructs were harvested after IR and then immunoprecipitated with anti-FLAG agarose, blots were probed with indicated antibodies. **c** Ubiquitinated CtIP was incubated with purified FLAG-WT USP52 or the S1003A mutant before or after IR to perform deubiquitination reaction assay in vitro, and then blotted with the indicated antibodies. **d, g** Control or USP52-depleted U2OS cells were transfected with indicated USP52 truncates before treated with 2 Gy IR for 2 h or 5 Gy IR for 5 h. RPA2 and Rad51 focus formation were detected by immunofluorescence (**d, f**) and quantified (**e, g**). Data are representative of three independent experiments. Each dot represents a single cell, and more than 200 cells were counted in each group for this experiment. Error bars represent SEM from this experiment. Scale bar, 10 μm. **h** Control or USP52-depleted HEK293T cells transfected with indicated USP52 constructs together with HR reporter were harvested for HR assay. Error bars represent SEM from three independent experiments. **i** USP52-depleted HEK293T cells were transfected with indicated USP52 constructs for 24 h before treated with or without IR. Harvested cells were then immunoprecipitated with nickel (His) beads and blotted with indicated antibodies. **j** Purified WT USP52 or the S1003A mutant was incubated with ATM for in vitro kinase assay, and then ATM-phosphorylated USP52 constructs were used for in vitro deubiquitination reaction with ubiquitinated GFP-CtIP. **k** Control or USP52-depleted HEK293T cells transfected with indicated USP52 constructs were harvested and immunoprecipitated with anti-FLAG agarose, and then subjected to blot with the indicated antibodies. **l, m** Control or USP52-depleted U2OS cells stably expressing indicated USP52 constructs were treated with IR or PARPi for 2 weeks. Cell viability was assessed using colony formation assay. Error bars represent SEM from three independent experiments.

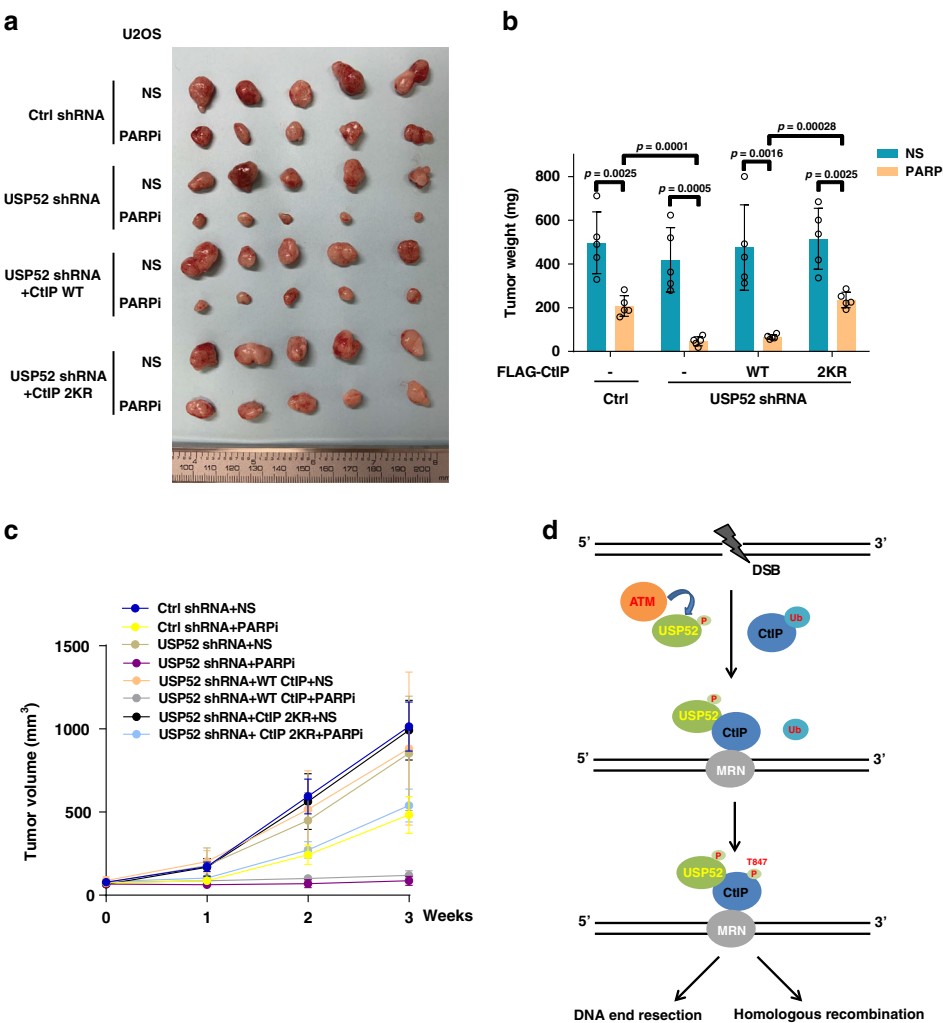

**Fig. 6 USP52–CtIP pathway is critical for enhancing chemotherapy of osteosarcoma cancer cells in vivo. a–c** Control or USP52-depleted U2OS cells stably expressing control vector, WT or 2KR mutant of CtIP were subcutaneously injected into the flank of NOD-SCID mice. Mice were treated with saline or olaparib (50 mg/kg i.p. 3 days × 6 times). Tumor images (**a**), tumor weight (**b**), and tumor volume (**c**) were then assessed. Data points in represent (mean ± SEM) are shown from n = 5 biologically independent samples by two-sided unpaired *t* test. **d** Working model of CtIP regulation by USP52.

how this process is controlled are still largely unknown. Here, we found that CtIP ubiquitination negatively regulates its phosphorylation at Thr-847. Furthermore, we discovered that CtIP is deubiquitinated by USP52, which removes the ubiquitin from CtIP to promote DNA end resection and HR repair (Fig. 6d). Our results suggest a critical role for ubiquitination in regulating CtIP activity and illustrate the regulatory mechanism of the USP52/CtIP pathway in the DDR.

USP52, also named PAN2 (poly(A) nuclease), is a bona fide DUB, which was reported to deubiquitinate and stabilize histone

chaperone ASF1A to facilitate chromatin assembly and breast carcinogenesis[34]. Though lacking an active-site cysteine residue, USP52 was able to hydrolyze K6-, K11-, K48-, K63-, and M1-linked ubiquitin chains through its UCH domain[34]. In addition, USP52 has been reported to be a key component of p-bodies which prevents the degradation of HIF1A mRNA and regulates HIF1A-mediated hypoxic response[44]. Here, we reveal that USP52 is also engaged in DNA end resection and HR repair through removing ubiquitin from CtIP, which dependent on the catalytic activity of the UCH domain. After DNA damage, CtIP is deubiquitinated in USP52-dependent manner, indicating that USP52/CtIP pathway plays a critical role in ensuring appropriate DNA end resection and HR repair. Based on the size of ubiquitiated CtIP, our data suggest that CtIP is not polyubiquitinated but rather is monoubiquitinated at K760 and K782. Because CtIP is a large protein, we could not detect an apparent shift caused by its ubiquitination. Ubiquitination can affect the protein stability, cellular localization, protein interactions or activity of target substrates[45,46]. We found that USP52-mediated CtIP deubiquitination does not affect CtIP protein level and recruitment to the DNA damage sites, but regulates the activity of CtIP through promoting the phosphorylation of CtIP at Thr-847. This increases DNA end resection and HR repair. Because the ubiquitination sites (K760 and K782) and the phosphorylation site (T847) are not the same (they are all located within the C-terminal domain of CtIP), it is possible that the ubiquitination of CtIP alters the conformation and subsequently masks the phosphorylation site. Alternatively, the ubiquitination of CtIP could be recognized by scaffold proteins or kinases/phosphatase with ubiquitin-binding domains that subsequently affect the phosphorylation of CtIP indirectly. The detailed mechanism regarding how CtIP ubiquitination affects its phosphorylation will need to be studied in further detail.

DUB activity and specificity are modulated by a variety of mechanisms including transcriptional and translational regulation, proteins interactions, and post-translational modifications to avoid inadvertent cleavage of non-substrate proteins[46,47]. Many DUBs activities are regulated by the phosphorylation on serine, threonine and tyrosine residues[46,47]. It was reported that a number of DUBs such as USP13, USP15, and UCHL3 were phosphorylated and activated by ATM in response to DNA damage to participate in the DNA repair process[29,30,48]. ATM is a core regulator in the DDR and phosphorylates hundreds of substrates containing closely spaced Ser-Gln (SQ) and Thr-Gln (TQ) motifs to orchestrate the response to DNA damage through initiating the regulation of cell cycle checkpoint, DNA repair and cell apoptosis[49–51]. Our results suggest that the phosphorylation of USP52 on Ser-1003 by ATM increases the activity of USP52 to deubiquitinate CtIP and promote DNA end resection and HR repair. Because the interaction between CtIP and USP52 was not changed before or after IR treatment, it is possible that the phosphorylation of USP52 increases its catalytic capability but not the affinity toward CtIP. Because USP52 protein level was not changed under IR treatment, it is also possible that USP52 might undergo conformational changes that are required for catalysis when phosphorylated by ATM.

PARP inhibitors are used routinely in the treatment of patients with HR-based DNA repair pathway deficits[52,53]. PARP inhibitors compete with NAD+ for the substrate binding to PARP, which converts the ssDNA breaks into toxic DSBs. Therefore, PARP inhibitors cause cell death in HR-defective cancer cells in a synthetic lethal manner[52,53]. CtIP is well-known to promote DSB repair through HR, and loss of CtIP markedly sensitizes cells to PARP inhibitors[18,38]. Therefore, targeting CtIP regulators is a potential strategy to enhance the efficiency of PARP inhibitors in cancer cells. We found that depletion of USP52 sensitives cells to

PARP inhibitor in CtIP deubiquitination dependent manner in vitro and in vivo, suggesting that USP52 might be a new therapeutic target for bone cancers which are resistant to standard PARP inhibitor therapy. Thus, future screening for USP52 enzymatic inhibitors which attenuate the DNA repair ability of CtIP in cancer cells could enhance the effects of PARP inhibitors.

## Methods

**Cell culture**. HEK293T and U2OS cell lines were purchased from ATCC. All of the cell lines have been tested and confirmed by the Mayo Clinic medical genome facility Center. U2OS ER-AsiSI cells were generated by Dr. Gaëlle Legube (Université de Toulouse, Toulouse, France). HEK293T cells were maintained in DMEM and U2OS cells were cultured with McCoy's 5A with 10% fetal bovine serum.

**Plasmids, reagents, and antibodies**. GFP-CtIP, full-length and truncated FLAG–CtIP were generously provided by Dr. Junjie Chen (MD Anderson Cancer Center, TX). FLAG–ASF1A was generously provided by Dr. Zhiguo Zhang (Columbia University, NY). Full-length and truncated FLAG–USP52 and Myc-USP52 were generously provided by Dr. Lei Shi (Tianjin Medical University, Tianjin, China). FLAG–USP52 S1003A and S469A, FLAG–CtIP K62R, K760R, K782R, and 2KR mutants were generated by site-directed mutagenesis (Stratagene). Flag–CtIP–T847E was purchased from Addgene. CHX, IgG agarose, anti-FLAG agarose, 3×FLAG peptide and ATM inhibitor KU55933 were purchased from Sigma Aldrich. Olaparib was purchased from Toronto Research Chemicals. Anti-USP52 (ab241505, 1:1000) was purchased from abcam; anti-CtIP (Thr847) (p1012-847, 1:2000), and anti-CtIP (Thr327) (p1012-327, 1:2000) were purchased from PhosphoSolutions; anti-RPA32 (sc-56770, 1:2000), anti-Ub (sc-8017, 1:2000) and anti-CtIP (sc-271339, 1:1000 for WB) were purchased from Santa Cruz; anti-FLAG (F1804, 1:2000) was purchased from Sigma; anti-pS345 Chk1 (2348, 1:1000), anti-Myc Tag (2276, 1:2000) and anti-SQ/TQ motif (9607, 1:1000) were purchased from CST; anti-CtIP (61141, 1:1000 for IF) was purchased from Active Motif; anti-GAPDH (60004-1-lg, 1:2000) was purchased from Proteintech; anti-Rad51 (GTX100469, 1:1000) was purchased from Genetex.

**RNA interference**. The following shRNAs from Sigma were used in this study: USP52 shRNA-1: 5′-CCTGCCTTCTTGCGCTTCATT-3′, USP52 shRNA-2: 5′-CAG TGATGATATTCGGCAGAT-3′, CtIP shRNA: 5′-CGGCAGCAGAATCT TAAATT-3′.

**Western blot and immunoprecipitation**. Cells were harvested and lysed with NETN buffer (20 mM Tris-HCl, pH 8.0, 100 mM NaCl, 1 mM EDTA, 0.5% Nonidet P-40 with 50 mM 10 mM NaF, and 1 mg per ml each of pepstatin A and aprotinin). After centrifugation at $12,000 \times g$ for 15 min, supernatant containing proteins was immunoprecipitated by incubating indicated antibodies or agarose beads overnight at 4 °C. The immunoprecipitates were washed with NETN and then centrifuged at $800 \times g$ for 1 min for three times. The immunoprecipitates were added with 50 μL 1× Laemmli buffer and then boiled for sodium dodecyl sulfate polyacrylamide gel electrophoresis separation, thereafter detected with indicated antibodies. All of uncropped blots are available in source data file.

**Denaturing Ni-NTA pull-down**. Cells were harvested and lysed in urea buffer composed of 8 M Urea, 0.1 M $NaH_2PO_4$, 30 mM NaCl and 0.01 M Tris (pH 8.0). Lysates were then sonicated to shear DNA and incubated with Ni-NTA agarose beads for 2 h at room temperature. After washing the beads with urea wash buffer (8 M Urea, 0.1 M $NaH_2PO_4$, 300 mM NaCl and 0.01 M Tris (pH 8.0)) for 5 times, the immunocomplexes were added with 1× Laemmli buffer and subjected to western blot.

**In vitro deubiquitination assays**. HEK293T cells were transfected with both His-Ub and GFP-CtIP for 36 h, ubiquitinated CtIP was then purified from the cell extracts with Ni-NTA agarose beads. U2OS cells stably expressing FLAG–USP52 and USP52 S1003A were in response to 10 Gy IR for 1 h, cells were then harvested for immunoprecipitation using FLAG agarose. Purified FLAG–USP52 and USP52 S1003A were eluted with FLAG-peptides (Sigma) according to manufacturer's instruction. For the in vitro deubiquitination assay, ubiquitinated CtIP protein was incubated with purified FLAG–USP52 and USP52 S1003A in deubiquitination buffer (50 mM Tris-HCl pH 8.0, 50 mM NaCl, 1 mM EDTA, 10 mM DTT, 5% glycerol) for 4 h at room temperature.

**Colony formation assay**. Totally, 1000 control or USP52-depleted U2OS cells stably expressing indicated constructs were plated in each well of 6-well plates and then treated with olaparib at indicated concentrations. After incubated for 12–14 days at 37 °C, colonies were stained with 5% GIEMSA and counted.

**Immunofluorescence staining**. U2OS cells were seeded on coverslips for 24 h before experiments. For RPA2 foci, cells were fixed with methanol: acetone (1:1) at

−20 °C for 20 min, while for Rad51 foci, cells were permeabilized for 10 min on ice with 0.1% Triton X-100 and then fixed by 3% paraformaldehyde for 10 min. Following this, coverslips were washed with phosphate-buffered saline (PBS) three times, cells were blocked with 5% goat serum for 30 min and then incubated with primary antibodies at 4 °C overnight. After washing with PBS, secondary antibody was added and incubated for 1 h at room temperature before stained with 4'6-diamidino-2-phenylindole (DAPI). Finally, the coverslips were mounted onto glass slides with anti-fade solution and visualized using a Nikon ECLIPSE E800 fluorescence microscope. The foci intensity was quantified with Image J software.

**HR assay**. Cells expressing indicated shRNAs or constructs were transfected with DR-GFP, pCBA-I-SceI, and pCherry. After 2 days cells were harvested and analyzed by fluorescence-activated flow cytometry (FACS) to examine the percentage of GFP-positive cells. Results were normalized to control group. The graphical account for FACS sequential gating/sorting strategies was provided in Supplementary Fig. 6.

**DNA resection measurement**. The percentage of ssDNA (ssDNA%) generated by resection was determined as previously described[54]. Briefly, ER-AsiSI U2OS cells expressing indicated shRNAs or constructs were treated with 1 μM 4-OHT for 4 h, cells were then harvested and genomic DNA was extracted with DNAzol reagent (Invitrogen) according to manufacturer's instruction. After that, 500 ng genomic DNA sample was digested or mock digested with BsrGI enzyme at 37 °C overnight. 2 μL DNA were used as templates in 25 μl of qPCR reaction containing 12.5 ml of 2× Taqman Universal PCR Master Mix (ABI), 0.5 mM of each primer and 0.2 mM probe. The sequences of qPCR primers and probes are shown in Supplementary Table 1. ΔCt was calculated from the Ct value of the digested sample subtracting the mock-digested sample. The ssDNA% was calculated with the following equation: $ssDNA\% = 1/(2^{(\triangle Ct-1)} + 0.5)*100$.

**Ub-AMC assay**. The in vitro DUB enzymatic assays using Ub-AMC as substrate was performed as described[55]. Briefly, 1 μL Ub-AMC (Boston Biochem) was performed in 50 μL reaction buffer (20 mM HEPES-KOH pH 7.8, 20 mM NaCl, 0.1 mg/mL ovalbumin, 0.5 mM EDTA, and 10 mM DTT) at room temperature, and then fluorescence at emission 460 nm and excitation 380 nm was monitored in an Infinite M1000 PRO Fluorometer (TECAN).

**Tumor xenograft**. Experiments were performed under the approval of the Institutional Animal Care and Use Committee at Mayo Clinic (Rochester, MN) under protocol A00002864-17. All mice used in this study were maintained under specific pathogen-free conditions, 21 ± 2 °C relative humidity of 45 ± 15%, and a 12-h light/dark cycle. Control or USP52-depleted U2OS cells expressing indicated constructs were injected subcutaneously into the flanks of 6-week-old female athymic nude Ncr nu/nu (National Cancer Institute/National Institutes of Health) mice. Each mouse was injected a 100 μl mixture of $2 \times 10^6$ cells with 50% growth factor reduced Matrigel (BD Biosciences). Mice bearing tumors of about 100 mm³ were divided into control group (saline) or PARPi group (50 mg kg⁻¹) and then intraperitoneally injected three times per week. Tumor volume was measured every week using calipers and calculated using the formula length × width². Mice were sacrificed for tumor dissection 3 weeks after the start of treatment.

**Statistics and reproducibility**. Data in bar and line graphs are presented as mean ± SEM of at least three independent experiments. All the Western blotting and micrograph data were repeated independently three times with similar results. For the animal xenograft study, data are presented as the mean ± SEM of five mice. Statistical analyses were performed in Microsoft Excel 2010 and GraphPad Prism7 with the Student's two-tailed t test. The flow cytometry data were gathered by Attune NxT Flow Cytometer software v2.6 and analyzed by flowjo V10.

**Reporting summary**. Further information on research design is available in the Nature Research Reporting Summary linked to this article.

## Data availability

Source data are provided with this paper. All data is available from the authors upon reasonable request. Source data are provided with this paper.

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

## Acknowledgements

This research was supported by funding from Center for Biomedical Discovery of Mayo Foundation. J.A.K. was supported by NIH T32 GM65841.

## Author contributions

M.G. and G.G. designed and conducted the experiments, analyzed the data, and wrote the paper; J.H., Y.C., J.A.K., F.Z., M.D., X.T., W.K., Q.Z., C.Z., K.L., and J.Y. provided technical and data analysis assistance. P.Y. provided the reagents for experiments. Z.L. conceived and supervised the project, designed the experiments, and analyzed the data.

## Competing interests

The authors declare no competing interests.
