## [Peer Review File · Nature Communications]

Reviewers' comments:

Reviewer #1 (Remarks to the Author):

The authors identified USP52 as a CtIP DUB that enforces DSB resection and HR. Although it remains unclear how CtIP deubiquitylation is coupled to its phosphorylation and activity, overall the study is clearly written and most conclusions are supported by sound and solid experimental evidence. I have only a few minor comments that the authors should address prior to formal acceptance of the manuscript for publication at Nat Commun.

1) Does USP52 deubiquitylate mono-ubiquitylated CtIP? In Discussion the authors mentioned that USP52 has been shown to hydrolyse a variety of ubiquitin chains. Since the authors proposed that CtIP may be mono-ubiquitylated it may be important to document that USP52 is active as a DUB in removing mono-ub adduct(s) from its substrates.

2) Is the USP52-CtIP interaction important for DSB resection and HR?

3) I suggest putting in an illustration to explain the DSB resection assay (Figure 2C).

4) U2OS ER-AsiSI cells were originally generated by the Gaelle Legube lab.

5) Figure 4H - Is p-CtIP(T847) blotting exogenous or endogenous CtIP? Ideally the blot should be done against Flag-IPed CtIP proteins.

6) Supp Figure 1 - Expression of some of the DUBs is difficult to make out. Better blots should be provided.

Reviewer #2 (Remarks to the Author):

In this paper by Lou and colleagues, the authors define a new regulatory modulation of DNA end resection and homologous recombination by the DUB USP52. Such regulation relies on the deubiquitination of the master resection regulator CtIP, but not by controlling protein levels but by affecting the key phosphorylation at T847. Moreover, such regulation is controlled by the DNA damage response, as USP52 is phosphorylated by ATM at S1003 and such post-translational modification is essential for USP52 role in resection and recombination. In general, the results showed by the authors are convincing and well controlled. They show and map the interaction between USP52 and CtIP. Also, genetically they can reconstitute the regulatory network, complementing USP52 depletion phenotypes with wt but not catalytic dead or S1003A mutants and also by expressing a non-ubiquitinated form of CtIP but not the wild type. I find the manuscript as it is interestingly and worthy to be published in Nature Communications.

There is only one thing that spike my curiosity. As the authors state in the discussion, CtIP is phosphorylated at T847 in a cell cycle dependent manner. However, their observations link this phosphorylation with a response to DNA damage via ATM. This is surprising, and as far as I know no one has before propose that T847 phosphorylation is DNA damage-dependent. Have the authors analyzed the effect of USP52 depletion on CtIP in different cell cycle stages? Does the ubiquitination of CtIP changes upon CDK inhibition? Could it be that the effect they are seeing depends on the accumulation of cells in S/G2 due to the ATM checkpoint? Finally, I think a nice experiment that will completely stablish the authors model is to express the CtIP-T847E mutant, that has been shown to overcome the cell cycle regulation of CtIP.

Additional points:

- In terms of CtIP accumulation at DNA damage sites, the untreated control is missing in all the

panels. This is important, as CtIP forms a punctuated pattern even without DNA damage. So, I am not completely convinced that the foci the authors see show DNA damage sites. A colocalization with a DSB marker is required, but my advice will be either use laser lines or ChIP with the AsiSI system in order to prove that the recruitment of CtIP is not affected.

Reviewer #3 (Remarks to the Author):

USP52 regulates DNA 1 end resection and chemosensitivity through removing inhibitory ubiquitination from CtIP
By Gao et al.

In this study of Gao et al., the authors study how CtIP, and by extension homologous recombination, is regulated by USP52-mediated deubiquitination. The authors demonstrate that USP52 interacts with CtIP, and define the required domains in either protein, and illustrate that the interaction is under control of DNA damage signalling. Lastly, the authors show that, similar to other HR-deficiencies, USP52 depletion sensitizes cells to PARP inhibition.

The manuscript is clear and the results are well-explained. However, the majority of the experiments follow a very similar framework for each figure, repeating a mostly fixed set of assays to study each new mutant introduced. This makes the results section somewhat stretched and repetitive. The observation that CtIP ubiquitylation is regulated by DNA damage signalling and this regulation is lost in USP52-depleted cells is intriguing, but hardly discussed and not sufficiently followed up on in relation to CtIP.

The observation that the HR defect that goes along with USP52 depletion leads to PARPi sensitivity is not very developed, and not surprising.

USP52 has been previously described to regulate the histone chaperone ASF1a. ASF1, in turn, has been linked to DNA repair through regulation of MDC1 (PMID: 28943310) and Rad51 (PMID: 29478807). These data are not discussed and could be a second mechanism that explains the observed DNA repair defects. In this light, it is important to test to which degree the observed effects can be linked to CtIP regulation.

While this manuscript provides a novel interaction between USP52 and CtIP, the experiments involving PARPi sensitization could be extended to strengthen the paper.

Specific comments:

1. In the introduction, it is mentioned that previous studies have reported that USP4 interacts with and promotes CtIP recruitment, but without affecting CtIP ubiquitination. While a panel of DUBs were used to screen for CtIP interactor, USP4 was not included in the panel and for subsequent experiments. USP4 should be included here.

2. It remains largely unclear how the regulation of CtIP by USP52 is controlled. Figure 1H shows that CtIP-ubiquitin levels are completely lost upon IR. This suggests that either the Ubiquitin ligase is inactivated by DDR signalling, or that DUB activity is increased. The observation that USP52 depletion completely abolished IR-induced loss of Ub-CtIP suggests that USP52 is activated. In Suppl. Figure S6, there is mentioning of a USP52 mutant that cannot be activated through phosphorylation by ATM. However, there is no data linking the effects of this mutant to CtIP. Yet, Figure 4H shows that in irradiated cells, also the phosphorylation of CtIP at the CDK site T847 is lost in USP52-depleted cells, suggesting that USP52 depletion also has other effects on CtIP. In conclusion, even though the authors show interaction between USP52-CtIP, and regulation of CtIP by USP, It remains largely unclear how this interaction works in conditions of DNA damage.

3. The majority of data investigate USP52/CtIP in conditions of IR, but the sensitivity to IR is not included in Figure 6. IR should be included in clonogenic survival assays.
4. Repeated experiments of Figure 1F-H, especially 1G: the claim of significantly reduced ubiquitination of CtIP with UCH-mutant, should be quantified based on multiple experiments
5. Figure 2D and F: intensity level of the staining of RPA and Rad51 foci seem to higher for shRNA control compared to USP52#1 and #2, even background levels seems increased shRNA control images. If foci intensity is really lower for USP52 depletion why is this not quantified together with reduced foci numbers?
6. For foci quantification: a weird cut-of is used (%positive cells with >10 foci per cell). Raw numbers of foci per cell should be quantified and displayed.
7. Legends for Figure 2D-G states 3 independent experiments are shown, however 2E shows 8 data points and 2G shows 7 data points per condition. Are these individual experiments or field of view? Please clarify or modify.
8. Text for Figure 2D-H on line 152-153 states 'depletion of USP52 significantly abrogated the formation of RPA and Rad51 foci and HR'. At best the data shows a 40% reduction in HR efficiency and a 50% reduction in cells with >10 foci, HR is clearly reduced but not abrogated by USP52 depletion. Again, quantification and visualization of data is problematic.
9. Figure 3: Why is FLAG-CtIP used in this experiment instead of blotting endogenous CtIP and p-CtIP (T847)? What time after irradiation were these cells harvested?
10. There is a clear inconsistency in the amount of irradiation used: In 3B-C 2Gy is used, in D-E 5Gy is used, in I 10Gy is used. Especially at higher doses of irradiation >5 Gy, HR components can become depleted and could interfere with HR function.
11. Figure 4: Line 199: Not potentially ubiquitinated, cited and curated mass spectrometric evidence
12. S Figure 5B K762R alone abolishes ubiquitination of CtIP, why was a double 2KR mutant required for Figure 4? Does the single K762R mutant alone not sufficiently stabilize HR function?
13. Figure 6: the reporting of data is very 'stretched'. The 4 panels could be incorporated in respective figures 2-5, as they are clearly extensions from each subset of shRNA or CtIP/USP52 mutant.

Minor points:

14. all the Suppl. figure titles contain the same typo: 'Supplementary' should be supplementary.
15. Figure 6E is explained in the results section as if these are data. This model should only be discussed in the discussion section.
16. Figure 5D shows Sigma sign at the mutation site instead of 'S'
17. Suppl Fig 1A: second blot: one USP lacks specific name (one but last lane).

Reviewers' comments:

Reviewer #1 (Remarks to the Author):

The authors identified USP52 as a CtIP DUB that enforces DSB resection and HR. Although it remains unclear how CtIP deubiquitylation is coupled to its phosphorylation and activity, overall the study is clearly written and most conclusions are supported by sound and solid experimental evidence. I have only a few minor comments that the authors should address prior to formal acceptance of the manuscript for publication at Nat Commun.

Thanks for the positive comments and suggestions.

1) Does USP52 deubiquitylate mono-ubiquitylated CtIP? In Discussion the authors mentioned that USP52 has been shown to hydrolyse a variety of ubiquitin chains. Since the authors proposed that CtIP may be mono-ubiquitylated it may be important to document that USP52 is active as a DUB in removing mono-ub adduct(s) from its substrates.

In the revised manuscript, we performed *in vitro* deubiquitination assays with ubiquitin-AMC (Ub-AMC), one of the most reliable artificial mono-Ub fluorescent DUB substrates, to detect whether USP52 is active as a DUB in removing mono-ub adduct(s) from its substrates. As shown in Figure S1H, WT USP52 but not USP52 Δ UCH was able to cleave Ub-AMC and release free AMC fluorescence, suggesting that the deubiquitinase activity of USP52 is responsible for hydrolyzing mono-ubiquitylated substrates. In addition, our *in vitro* results in Figure 5C and H also show that monoubiquitylated CtIP (according to the size of ubiquitylated CtIP) could be cleaved by purified USP52, indicating that USP52 is able to remove mono-ubiquitin from its substrate CtIP.

2) Is the USP52-CtIP interaction important for DSB resection and HR?

We explored whether the USP52-CtIP interaction is important for DSB resection and HR. As shown in Figure S2E and 2H-I, WT USP52, but not the USP52 Δ WD40 truncate, successfully rescued DSB resection and HR efficiency in USP52 depleted cells, suggesting that the USP52-CtIP interaction is important for the role of USP52 in DSB resection and HR. This corroborates our finding that WD40 domain is required for the interaction between USP52 and CtIP.

S2E

2H

2I

3) I suggest putting in an illustration to explain the DSB resection assay (Figure 2C).

In the revised manuscript, we have added an illustration to explain the DSB resection assay system, as shown in Figure 2B.

4) U2OS ER-AsiSI cells were originally generated by the Gaelle Legube lab.

We have added the information in the section of “Methods” “U2OS ER-AsiSI cells were generated by Dr Gaëlle Legube (Université Toulouse, Toulouse, France).

5) Figure 4H - Is p-CtIP(T847) blotting exogenous or endogenous CtIP? Ideally the blot should be done against Flag-IPed CtIP proteins.

In the revised manuscript, we have repeated this experiment to detect p-CtIP (T847) in Flag-IPed cell lysates. The results are shown in Figure 4F.

6) Supp Figure 1 - Expression of some of the DUBs is difficult to make out. Better blots should be provided.

In the revised manuscript, we have repeated this experiment to ensure every DUB is expressed well and then detected their interaction with CtIP. The results are shown in Figure S1.

Reviewer #2 (Remarks to the Author):

In this paper by Lou and colleagues, the authors define a new regulatory modulation of DNA end resection and homologous recombination by the DUB USP52. Such regulation relies on the deubiquitination of the master resection regulator CtIP, but not by controlling protein levels but by affecting the key phosphorylation at T847. Moreover, such regulation is controlled by the DNA damage response, as USP52 is phosphorylated by ATM at S1003 and such post-translational modification is essential for USP52 role in resection and recombination. In general, the results showed by the authors are convincing and well controlled. They show and map the interaction between USP52 and CtIP. Also, genetically they can reconstitute the regulatory network, complementing USP52 depletion phenotypes with wt but not catalytic dead or S1003A mutants and also by expressing a non-ubiquitinated form of CtIP but not the wild type. I find the manuscript as it is interestingly and worthy to be published in Nature Communications.

There is only one thing that spike my curiosity. As the authors state in the discussion, CtIP is phosphorylated at T847 in a cell cycle dependent manner. However, their observations link this phosphorylation with a response to DNA damage via ATM. This is surprising, and as far as I know no one has before propose that T847 phosphorylation is DNA damage-dependent. Have the authors analyzed the effect of USP52 depletion on CtIP in different cell cycle stages? Does the ubiquitination of CtIP changes upon CDK inhibition? Could it be that the effect they are seeing depends on the accumulation of cells in S/G2 due to the ATM checkpoint? Finally, I think a nice experiment that will completely stablish the authors model is to express the CtIP-T847E mutant, that has been shown to overcome the cell cycle regulation of CtIP.

Thanks for the positive comments and suggestions. Dr Markus Löbrich's group generated a phosphospecific antibody for detecting CtIP-T847 phosphorylation and found that CtIP-T847 phosphorylation level sharply peaked at 2 h after IR treatment and was weakly increased at earlier time points ¹. As shown in Figure S3D, our results also show that IR-induced CtIP-T847 phosphorylation was increased between 0.5 h to 2h in response to IR, which is consistent with their conclusion, although the peak time of CtIP-T847 phosphorylation is a little different.

To test whether USP52-regulated CtIP deubiquitination is cell cycle dependent, we detected the ubiquitination levels of CtIP at different cell cycle stages. As shown in Figure S3F, the basal CtIP ubiquitination levels were similar in different cell cycle stages, and CtIP ubiquitination levels were all increased in different cell cycle phases of USP52-depleted cells. This suggests that CtIP ubiquitination regulated by USP52 is not cell cycle dependent. In addition, we compared the ubiquitination levels of WT CtIP and the CtIP-T847E mutant, which is able to overcome the cell cycle regulation of CtIP under IR treatment, and found that WT CtIP and

CtIP-T847E mutant ubiquitination levels were both significantly decreased after IR treatment (Figure S3G). Moreover, CDK1/2 kinase inhibition by BMS-265246 also had no effect on IR-induced CtIP deubiquitination (Figure S3H), further suggesting that CtIP ubiquitination/deubiquitination is not cell cycle dependent.

Additional points:

- In terms of CtIP accumulation at DNA damage sites, the untreated control is missing in all the panels. This is important, as CtIP forms a punctuated pattern even without DNA damage. So, I am not completely convinced that the foci the authors see show DNA damage sites. A colocalization with a DSB marker is required, but my advice will be either use laser lines or ChIP with the AsiSI system in order to prove that the recruitment of CtIP is not affected.

In the revised manuscript, we have added the untreated control for CtIP foci in all the panels. As shown in Figure 2A, the focus formation of CtIP was significantly increased when treated with IR, but USP52 depletion had no effect on CtIP foci formation before or after IR treatment. In addition, we also used ChIP assay with the AsiSI system to detect whether the DNA damage sites recruitment of CtIP is affected by USP52. As shown in Figure 2C, the recruitment of FLAG-CtIP to DNA damage sites was significantly increased when cells were treated with 4-OHT, but there was almost no difference between control and USP52 depleted cells, suggesting that USP52 doesn't affect the DNA damage sites recruitment of CtIP.

Reviewer #3 (Remarks to the Author):

USP52 regulates DNA 1 end resection and chemosensitivity through removing inhibitory ubiquitination from CtIP

By Gao et al.

In this study of Gao et al., the authors study how CtIP, and by extension homologous recombination, is regulated by USP52-mediated deubiquitination. The authors demonstrate that USP52 interacts with CtIP, and define the required domains in either protein, and illustrate that the interaction is under control of DNA damage signalling. Lastly, the authors show that, similar to other HR-deficiencies, USP52 depletion sensitizes cells to PARP inhibition.

The manuscript is clear and the results are well-explained. However, the majority of the experiments follow a very similar framework for each figure, repeating a mostly fixed set of assays to study each new mutant introduced. This makes the results section somewhat stretched and repetitive. The observation that CTIP ubiquitylation is regulated by DNA damage signalling and this regulation is lost in USP52-depleted cells is intriguing, but hardly discussed and not sufficiently followed up on in relation to CtIP.

The observation that the HR defect that goes along with USP52 depletion leads to PARPi sensitivity is not very developed, and not surprising.

USP52 has been previously described to regulate the histone chaperone ASF1a. ASF1, in turn, has been linked to DNA repair through regulation of MDC1 (PMID: 28943310) and Rad51 (PMID: 29478807). These data are not discussed and could be a second mechanisms that explains the observed DNA repair defects. In this light, it is important to test to which degree the observed effects can be linked to CtIP regulation.

Thank you for the positive comments and suggestions. In the revised manuscript, we detected whether ASF1A, another known substrate of USP52, was also involved in DSB resection and HR. Because USP52 stabilizes ASF1A, if the observed effects are partly due to ASF1A, we would expect that ectopic expression of ASF1A would rescue some of the defects. As shown in Figure S2F-H, ectopic expression of ASF1A couldn't rescue USP52 depletion-induced defects in DSB resection and HR, indicating that ASF1A is not involved in USP52-mediated DSB resection and HR repair. In contrast, CtIP-2KR expression was able to rescue the defects caused by USP52 depletion (Figure 4B). It is possible that ASF1A might be participated in other aspects of USP52-mediated DNA damage response, such as replication stress response and cell survival.

F

G

H

While this manuscript provides a novel interaction between USP52 and CtIP, the experiments involving PARPi sensitization could be extended to strengthen the paper.

Thank you for the suggestion. To further strengthen the role of USP52-CtIP pathway involved in PARPi sensitization, we investigated the relationship between USP52 and CtIP in response to PARPi *in vivo*. As shown in Figure 6A-C, USP52 depletion enhanced the sensitive to PARPi in an osteosarcoma xenograft model, which could be reversed by the overexpression of the 2KR mutant, but not WT CtIP. Taken together, our results suggest that the USP52-CtIP pathway may be a potential therapeutic target of cancer cells to PARPi.

Specific comments:

1. In the introduction, it is mentioned that previous studies have reported that USP4 interacts with and promotes CtIP recruitment, but without affecting CtIP ubiquitination. While a panel of DUBs were used to screen for CtIP interactor, USP4 was not included in the panel and for subsequent experiments. USP4 should be included here.

We added USP4 in the panel and found that both USP52 and USP4 interacted with CtIP (Figure S1). The results were also shown in review 1 question 6.

2. It remains largely unclear how the regulation of CtIP by USP52 is controlled. Figure 1H shows that CtIP-ubiquitin levels are completely lost upon IR. This suggests that either the Ubiquitin ligase is inactivated by DDR signaling, or that DUB activity is increased. The observation that USP52 depletion completely abolished IR-induced loss of Ub-CtIP suggests that USP52 is activated. In Suppl. Figure S6, there is mentioning of a USP52 mutant that cannot be activated through phosphorylation by ATM. However, there is no data linking the effects of this mutant to CtIP.

Thank you for pointing this out. To further examine whether the phosphorylation of USP52 on S1003 regulates its role in CtIP deubiquitination, we reconstituted WT or the S1003A mutant of USP52 in USP52-deficient cells and detected the ubiquitination level of CtIP with or without IR treatment. As shown in Figure 5G, CtIP ubiquitination was significantly decreased in cells expressing WT but not the S1003A mutant of USP52 after IR treatment, indicating that USP52 phosphorylation is critical for its role in deubiquitinating CtIP following IR exposure. In addition, we performed an *in vitro* kinase reaction of USP52 by ATM followed by an *in vitro* deubiquitination assay to further confirm whether USP52 phosphorylation activates its activity. As shown in Figure 5H, WT but not S1003A mutant of USP52, after incubating with ATM,

could further decrease the ubiquitination level of CtIP, suggesting that ATM-mediated USP52 phosphorylation is important for USP52 activation to deubiquitinate CtIP following DNA damage.

G

H

Yet, Figure 4H shows that in irradiated cells, also the phosphorylation of CtIP at the CDK site T847 is lost in USP52-depleted cells, suggesting that USP52 depletion also has other effects on CtIP.

Yes, our results showed that CtIP 2KR but not WT CtIP rescued USP52 depletion-reduced CtIP-T847 phosphorylation under IR treatment, indicating that the USP52-CtIP axis regulates CtIP phosphorylation at T847 (which has been shown to be critical for CtIP activation) following DNA damage. Because the ubiquitination sites (K760 and K782) and the phosphorylation site (T847) are not the same (they all located within the C-terminal domain of CtIP), it is possible that the ubiquitination of CtIP alters the structure of CtIP protein and subsequently masks the phosphorylation site, or the ubiquitination of CtIP is recognized by scaffold proteins or kinases/phosphatase which have ubiquitin-binding domains, which subsequently affects the phosphorylation of CtIP indirectly. The detailed mechanism for how CtIP Ub affects its phosphorylation will need to be studied in further detail.

Even though the authors show interaction between USP52-CtIP, and regulation of CtIP by USP, It remains largely unclear how this interaction works in conditions of DNA damage.

In the revised manuscript, we studied the role of USP52-CtIP interaction in resection and HR. The results are also shown in review 1 question 2.

3. The majority of data investigate USP52/CtIP in conditions of IR, but the sensitivity to IR is not included in Figure 6. IR should be included in clonogenic survival assays.

In the revised manuscript, we detected whether USP52/CtIP pathway also affects the sensitivity of cells to IR treatment. As shown in Figure 2J, 3H and 5J, depletion of USP52 sensitized cells to IR treatment, and reconstitution with WT USP52 but not USP52 ΔUCH or USP52 S1003A reversed USP52 depletion-induced IR hypersensitivity. Meanwhile, as shown in Figure 4G, expression of CtIP 2KR but not WT CtIP in USP52 depleted cells reversed IR-

induced hypersensitivity. Taken together, these results indicate that USP52/CtIP pathway plays an important role in regulating IR sensitivity.

2J

3H

4G

5J

4. Repeated experiments of Figure 1F-H, especially 1G: the claim of significantly reduced ubiquitination of CtIP with UCH-mutant, should be quantified based on multiple experiments

In the revised manuscript, we quantified the intensities of the bands from the three independent experiments (Figure 1F-H, S1D-G and S1I-J) using image J, the results are shown in Figure 1F-H. “Depletion of USP52 increased ubiquitin conjugates of CtIP about 3-fold compared to control. Meanwhile, re-expression of USP52 removed about 75% of ubiquitin from CtIP, whereas reconstitution of USP52 ΔUCH had almost no effect on the ubiquitination of CtIP, indicating that USP52 enzymatic activity is essential for regulating CtIP deubiquitination. In addition, about eighty percent of the ubiquitination level of CtIP was reduced when treated with IR, suggesting that DNA damage-induced CtIP deubiquitination occurs in cells. However, in USP52-depleted cells, this phenomenon was almost totally blocked. Taken together, these results suggest that USP52 is a DUB that regulates CtIP deubiquitination under physiological conditions and following DNA damage.

5. Figure 2D and F: intensity level of the staining of RPA and Rad51 foci seem to higher for shRNA control compared to USP52#1 and #2, even background levels seems increased shRNA control images. If foci intensity is really lower for USP52 depletion why is this not quantified together with reduced foci numbers?

In the revised manuscript, we examined the effect of USP52 on the intensity of RPA2 and Rad51 foci, and found that in USP52-depleted cells, the intensity of RPA2 and Rad51 foci were also weaker compared to control cells, indicating that both the number and intensity of RPA2 and Rad51 foci were decreased when USP52 is depleted.

6. For foci quantification: a weird cut-of is used (%positive cells with >10 foci per cell). Raw numbers of foci per cell should be quantified and displayed.

Thank you for the suggestion. In the revised manuscript, we recalculated the raw numbers of foci per cell for foci quantification, and the results are shown in Figure 2E-F, 3B-C, 4C-D and 5D-E. The RPA2 and Rad51 foci were significantly decreased when USP52 was depleted (2E-F), whereas reconstitution of WT USP52 but not USP52 Δ UCH (3B-C) or USP52 S1003A (5D-E) in USP52-deficient cells could restore RPA2 and Rad51 foci. Moreover, overexpression of CtIP 2KR but not WT CtIP in USP52-depleted cells reversed USP52 depletion-reduced RPA2 and Rad51 foci (4C-D).

7. Legends for Figure 2D-G states 3 independent experiments are shown, however 2E shows 8 data points and 2G shows 7 data points per condition. Are these individual experiments or field of view? Please clarify or modify.

In the revised manuscript, we recalculated the raw numbers of foci per cell for foci quantification, and the results are shown in question 6, and we also changed the statement to “Representative images and quantification of RPA2 foci in control or USP52-depleted U2OS cells when treated with 2 Gy IR for 2 h. Data are representative of three independent experiments. Each dot represents a single cell, and more than 200 cells were counted in each group for this experiment. Error bars represent SEM from this experiment”.

8. Text for Figure 2D-H on line 152-153 states ‘depletion of USP52 significantly abrogated the formation of RPA and Rad51 foci and HR’. At best the data shows a 40% reduction in HR efficiency and a 50% reduction in cells with >10 foci, HR is clearly reduced but not abrogated by USP52 depletion. Again, quantification and visualization of data is problematic.

In the revised manuscript, we changed this statement to “depletion of USP52 significantly decreased the formation of RPA and Rad51 foci and HR” and we also recalculated the raw numbers of foci per cell for foci quantification, and the results were shown in question 6.

9. Figure 3: Why is FLAG-CtIP used in this experiment instead of blotting endogenous CtIP and

p-CtIP (T847)? What time after irradiation were these cells harvested?

In the revised manuscript, we detected the phosphorylation of endogenous CtIP at T847 after IR treatment. As shown in the response to reviewer 2, the peak of p-CtIP T847 appeared between 0.5 h to 2 h after treated with 5 Gy IR, which consisted with the previous report that IR-induced a transient phosphorylation of CtIP-T847; therefore, we chose 1 h after 5 Gy IR treatment to detect the effect of USP52 on endogenous CtIP-T847 phosphorylation. As shown in Figure 3G, USP52 depletion significantly decreased IR-induced CtIP-T847 phosphorylation, whereas reconstitution of WT USP52 but not USP52 Δ UCH into USP52-depleted cells effectively reversed the reduction of CtIP-T847 phosphorylation.

10. There is a clear inconsistency in the amount of irradiation used: In 3B-C 2Gy is used, in D-E 5Gy is used, in I 10Gy is used. Especially at higher dosed of irradiation >5 Gy, HR components can become depleted and could interfere with HR function.

We used 2 Gy detect RPA foci^{2,3}. In our hand, 2 Gy did not give strong signal for Rad51, therefore, we used a relative higher dose of IR (5Gy) to observe a more visible Rad51 foci, which was also used by another researchers^{4,5,6}. In addition, in the revised manuscript, we have repeated Figure 3I and lowered the dose to 5 Gy according to the advice.

11. Figure 4: Line 199: Not potentially ubiquitinated, cited and curated mass spectrometric evidence

In the revised manuscript, we added the mass spectrometric evidences for CtIP ubiquitination. As shown below, “previous mass spectrometric data showed that the K62, K314, K360, k378, K410, K438, K526, K530, K585, K604, K613, K640, K760 and K782 were ubiquitinated sites of CtIP by using UbiSite or SEPTM strategy”^{7,8}. Among them, K62, K760 and K782 were located at the N and C-terminals of CtIP.

12. S Figure 5B K762R alone abolishes ubiquitination of CtIP, why was a double 2KR mutant required for Figure 4? Does the single K762R mutant alone not sufficiently stabilize HR function?

In the longer exposure, K762R still has residual ubiquitination (Figure S4B). We further investigated whether the K782R mutant of CtIP alone was sufficient to compensate USP52 depletion-reduced resection and HR repair. We reconstituted control and USP52-deficient cells

with CtIP K782R or CtIP 2KR mutants. As shown in Figure S4C-D, the CtIP 2KR mutant almost totally while the CtIP K782R mutant only partially restored USP52 depletion-reduced resection and HR repair, suggesting that deubiquitination of both K760 and K782 sites of CtIP are indispensable for the role of CtIP in resection and HR repair.

13. Figure6: the reporting of data is very ‘stretched’. The 4 panels could be incorporated in respective figures 2-5, as they are clearly extensions from each subset of shRNA or CtIP/USP52 mutant.

Thank you for the suggestion. In the revised manuscript, we have incorporated the four panels into Figure 2-5, respectively.

Minor points:

14. all the Suppl. figure titles contain the same typo: ‘Supplimentary’ should be supplementary.

Thanks for you to point out the errors here and below. In the revised manuscript, we have corrected all the ‘Supplimentary’ to “Supplementary”

15. Figure 6E is explained in the results section as if these are data. This model should only be discussed in the

In the revised manuscript, we described the model in the discussion section.

16. Figure 5D shows Sigma sign at the mutation site instead of ‘S’

We have corrected Σ to S.

17. Suppl Fig 1A: second blot: one USP lacks specific name (one but last lane).

We have added the name USP7 in this lane.

Reference

1. Barton O, *et al.* Polo-like kinase 3 regulates CtIP during DNA double-strand break repair in G1. *The Journal of cell biology* **206**, 877-894 (2014).
2. Bakr A, *et al.* Involvement of ATM in homologous recombination after end resection and RAD51 nucleofilament formation. *Nucleic acids research* **43**, 3154-3166 (2015).
3. Ha K, *et al.* The anaphase promoting complex impacts repair choice by protecting ubiquitin signalling at DNA damage sites. *Nature communications* **8**, 15751 (2017).
4. Raderschall E, Golub EI, Haaf T. Nuclear foci of mammalian recombination proteins are located at single-stranded DNA regions formed after DNA damage. *Proceedings of the National Academy of Sciences of the United States of America* **96**, 1921-1926 (1999).
5. Sak A, Stueben G, Groneberg M, Bocker W, Stuschke M. Targeting of Rad51-dependent homologous recombination: implications for the radiation sensitivity of human lung cancer cell lines. *British journal of cancer* **92**, 1089-1097 (2005).
6. Agathangelou A, *et al.* USP7 inhibition alters homologous recombination repair and targets CLL cells independently of ATM/p53 functional status. *Blood* **130**, 156-166 (2017).
7. Akimov V, *et al.* UbiSite approach for comprehensive mapping of lysine and N-terminal ubiquitination sites. *Nature structural & molecular biology* **25**, 631-640 (2018).
8. Mertins P, *et al.* Integrated proteomic analysis of post-translational modifications by serial enrichment. *Nature methods* **10**, 634-637 (2013).

REVIEWERS' COMMENTS

Reviewer #1 (Remarks to the Author):

The authors have satisfactorily addressed all my concerns.

Reviewer #2 (Remarks to the Author):

As mentioned in my first review, I think the paper is interesting and well performed. The authors have satisfied my curiosity performing the appropriate experiments and controls. Therefore, I am happy to endorse its publication.

Reviewer #3 (Remarks to the Author):

The authors have adequately responded to my comments, and I support publication of their revised manuscript.